# Association Between Low Omega-6 Polyunsaturated Fatty Acid Levels and the Development of Delirium in the Coronary Intensive Care Unit

**DOI:** 10.3390/nu17121979

**Published:** 2025-06-11

**Authors:** Yurina Sugita-Yamaguchi, Tetsuro Miyazaki, Kazunori Shimada, Megumi Shimizu, Shohei Ouchi, Tatsuro Aikawa, Tomoyuki Shiozawa, Kiyoshi Takasu, Masaru Hiki, Shuhei Takahashi, Katsuhiko Sumiyoshi, Tohru Minamino

**Affiliations:** 1Department of Biology and Cardiovascular Medicine, Graduate School of Medicine, Juntendo University, 2-1-1 Hongo, Bunkyo 113-8421, Japan; yrsugita@juntendo.ac.jp (Y.S.-Y.); shimakaz@juntendo.ac.jp (K.S.); megumi-s@juntendo.ac.jp (M.S.); uchi@juntendo.ac.jp (S.O.); taikawa@juntendo.ac.jp (T.A.); t-shio@juntendo.ac.jp (T.S.); ktakasu@juntendo.ac.jp (K.T.); ma-hiki@juntendo.ac.jp (M.H.); syutaka@juntendo.ac.jp (S.T.); t.minamino@juntendo.ac.jp (T.M.); 2Laboratory of Bioregulatory Clinical Pharmacology, Faculty of Pharmacy, Juntendo University, 6-8-1 Hinode, Urayasu 279-0013, Japan; 3Department of Health and Nutrition, Faculty of Human Sciences, Tokiwa University, 1-430-1 Miwa, Mito 310-5385, Japan; kazus@tokiwa.ac.jp

**Keywords:** omega-6 PUFAs, delta-5 desaturase, delirium, cardiovascular disease, cardiac intensive care unit, preventive medicine, aging

## Abstract

**Background:** Delirium is frequently observed in patients admitted to the intensive care unit, and is associated with mortality and morbidity. Although several studies have reported an association between polyunsaturated fatty acids (PUFAs) and cognitive disorders, the association between PUFA levels and development of delirium in patients with acute cardiovascular disease remains unknown. **Objective:** This study aimed to clarify the association between PUFA levels and development of delirium in the coronary intensive care unit (CICU). **Methods:** We enrolled 590 consecutive patients (mean age, 70 ± 14 years) admitted to the CICU of Juntendo University Hospital. Fasting serum PUFA levels were measured within 24 h of admission. Delta-5 desaturase activity was estimated as the ratio of arachidonic acid (AA) to dihomo-gamma-linolenic acid (DGLA). Furthermore, delirium was defined as patients having a delirium score of ≥4 using the Intensive Care Delirium Screening Checklist. **Results:** Delirium was observed in 55 patients. DGLA levels were significantly lower, and delta-5 desaturase activity was significantly higher in patients with delirium than in those without delirium (both *p* < 0.001). Conversely, AA alone and omega-3 PUFAs did not differ between the groups. Additionally, DGLA and AA levels, but not omega-3 PUFA levels, were negatively associated; delta-5 desaturase activity was positively associated with the delirium score (both *p* < 0.001). The duration of delirium was significantly associated with DGLA and AA levels (*p* = 0.001 and *p* = 0.004, respectively). Moreover, multivariate analysis showed that decreased DGLA and increased delta-5 desaturase activity remained significant predictors of delirium. **Conclusions:** Low omega-6 PUFA levels and high delta-5 desaturase activity on admission were significantly associated with the development of delirium in the CICU, indicating that the evaluation of low omega-6 PUFA levels and related enzymes may identify patients at a high risk of developing delirium.

## 1. Introduction

Delirium frequently occurs in patients receiving intensive care. Approximately one-third of hospitalized individuals aged 70 years or older experience this condition [1]. Numerous risk factors have been linked to the onset of delirium, including advanced age, impaired ability to perform daily activities, pre-existing dementia, the administration of antipsychotic medications, use of feeding tubes, peripheral venous or urinary catheters, physical restraints, and nutritional deficiencies [2,3]. In particular, inadequate nutritional intake has been implicated in the early emergence of postoperative delirium among surgical patients [4,5]. Moreover, our prior research demonstrated that malnutrition contributes to the development of delirium in individuals with acute cardiovascular conditions admitted to a coronary intensive care unit (CICU) [6]. However, the pathophysiological mechanisms underlying delirium remain poorly understood [1]. Delirium has been linked to adverse clinical outcomes, including cognitive and functional decline, an increased likelihood of discharge to care facilities rather than returning home, and both short- and long-term mortality—especially among patients with acute cardiac conditions [2,7]. Therefore, early identification and prevention of delirium are essential for improving prognosis in this vulnerable population.

Essential fatty acids (EFAs) and their long-chain polyunsaturated derivatives (LCPs) play a fundamental role in human growth and health maintenance. Since humans lack the ability to produce EFAs and can only inefficiently convert them into LCPs, daily dietary intake of EFAs is essential [8,9]. The primary EFAs in the *n*-6 and *n*-3 families are linoleic acid (LA; 18:2*n*-6) and alpha-linolenic acid (ALA; 18:3*n*-3), respectively. These parent fatty acids undergo desaturation and elongation in the body, leading to the formation of longer-chain, more unsaturated LCPs. Among the metabolites of LA, dihomo-gamma-linolenic acid (DGLA; 20:3*n*-6) and arachidonic acid (AA; 20:4*n*-6) are particularly notable. These are classified as omega-6 polyunsaturated fatty acids (PUFAs) due to the location of their first double bond at the sixth carbon from the methyl end. On the other hand, eicosapentaenoic acid (EPA; 20:5*n*-3) and docosahexaenoic acid (DHA; 22:6*n*-3), which are generated from ALA, belong to the omega-3 PUFA family, as their first double bond appears at the third carbon [10,11]. Omega-3 PUFAs are predominantly found in marine oils, such as fish oil, whereas omega-6 PUFAs are abundant in meats and in plant oils like sunflower, safflower, and corn oil [12].

In recent years, nutritional assessment has become an increasingly recognized component of cardiovascular disease management [13]. Of particular interest is the role of PUFAs in cardiovascular health, which has been the subject of extensive investigation [14,15,16,17,18]. In our earlier research, we found that reduced levels of omega-6 polyunsaturated fatty acids were significantly linked to increased long-term mortality across different nutritional statuses in patients with acute decompensated heart failure (ADHF) [19]. There is still a lack of evidence identifying which types of PUFAs may help reduce the incidence of delirium and improve prognosis in individuals with heart disease.

Accordingly, this study sought to investigate the relationship between the polyunsaturated fatty acid profile at the time of admission and the onset of delirium in patients presenting with acute cardiovascular conditions.

## 2. Materials and Methods

### 2.1. Study Design and Participants

This investigation was embedded within an ongoing prospective cohort study focusing on biomarkers in CICU-admitted patients. Although the hypothesis concerning PUFAs emerged retrospectively, all data were gathered in a systematic and prospective manner. We screened 783 consecutive patients who were admitted to the CICU of Juntendo University Hospital between January 2015 and December 2016. Among patients with multiple CICU admissions during the study period, only the first admission was included; subsequent admissions (*n* = 85) were excluded. An additional 36 patients were excluded due to incomplete delirium score assessments. Because of the difficulty in evaluating delirium, 9 patients who either required mechanical ventilation or exhibited profound impairments in consciousness were also excluded. Furthermore, 63 patients without PUFA measurements were excluded. As a result, 590 patients with both completed delirium assessments and PUFA level measurements were included in the final analysis. These patients were divided into two groups—delirium and non-delirium—to investigate the association between PUFA levels and the occurrence of delirium.

Delirium was identified based on an Intensive Care Delirium Screening Checklist (ICDSC) score of 4 or higher [20]. The evaluations were performed three times daily by independent nursing staff. To support the circadian rhythm, patients in the CICU were exposed to natural daylight during the day, and the lights were dimmed at night to simulate a normal day–night cycle. Additionally, to help orient patients to time and place, each room was equipped with a clock, and nurses routinely confirmed the date with patients each day. Bringing photographs of family members was also permitted.

All relevant data were collected from medical records at the time of hospital admission, including age, sex, body mass index (BMI), left ventricular ejection fraction (LVEF), comorbidity history (e.g., diabetes mellitus, dyslipidemia, hypertension, atrial fibrillation, dementia, cerebral infarction, and malignancy), reason for admission (e.g., ADHF, acute coronary syndrome (ACS), aortic dissection, pulmonary thromboembolism/deep vein thrombosis, ventricular tachycardia/ventricular fibrillation, and other causes), laboratory test results, and pre-admission medication use.

To ensure consistency and reproducibility, the definitions of key clinical conditions at the time of hospital admission used in this study are provided as follows: BMI was measured at the time of admission. LVEF was measured by transthoracic echocardiography upon admission. Diabetes mellitus was determined by either a documented diagnosis in the admission records, a hemoglobin A1c (HbA1c) level exceeding 6.5% as standardized by the National Glycohemoglobin Standardization Program, or the use of antidiabetic medications, including insulin. Dyslipidemia was identified through a recorded diagnosis at admission, abnormal lipid levels—specifically, triglycerides (TG) > 150 mg/dL, low-density lipoprotein cholesterol (LDL-C) > 140 mg/dL, or high-density lipoprotein cholesterol (HDL-C) < 40 mg/dL—or current use of lipid-lowering therapy. Hypertension was defined as a diagnosis noted at admission, systolic blood pressure greater than 140 mmHg and/or diastolic pressure over 90 mmHg, or ongoing treatment with antihypertensive drugs. Atrial fibrillation, dementia, prior cerebral infarction, and malignancy were confirmed based on documented admission diagnoses in the medical records.

The definitions of reasons for hospital admission were as follows: The diagnosis of ADHF was based on the criteria established by the Framingham study. [21]. ACS was characterized by the presence of unstable angina, non–ST-elevation myocardial infarction, or ST-elevation myocardial infarction [22]. Aortic diseases, pulmonary thromboembolism/deep vein thrombosis, ventricular tachycardia/ventricular fibrillation, and other conditions were defined based on the documented admission diagnoses in the medical records.

All subjects provided informed consent, and this study was approved by the Ethical Committee of Juntendo University Hospital.

### 2.2. Blood Sampling

Serum levels of PUFAs were assessed under fasting conditions within 24 h after hospital admission. Blood samples were withdrawn into ethylenediamine tetra acetic acid vacutainers and centrifuged at 2000 rpm at 4 °C for 20 min. Serum samples were stored at −80 °C until for PUFA concentration analysis. PUFAs, including DGLA, AA, EPA, and DHA, were measured using gas chromatography by SRL, Inc. (Tokyo, Japan), a certified clinical laboratory. All other laboratory tests were performed at the hospital’s central laboratory using standardized procedures. Plasma concentrations of total cholesterol (TC), TG, and HDL-C were determined using conventional enzymatic assays, while LDL-C was estimated via the Friedewald equation. Additional laboratory parameters, including albumin, HbA1c, C-reactive protein (CRP), creatinine, and *N*-terminal pro-brain natriuretic peptide (NT-proBNP), were measured according to standardized clinical protocols [23].

### 2.3. Statistical Analysis

Continuous data are presented as means with standard deviations, while categorical data are shown as percentages. Group comparisons were conducted using the Student’s t-test, the chi-square test, or Fisher’s exact test, as appropriate. The distribution of continuous variables was evaluated, and normality was assessed using the Shapiro–Wilk test. For variables that did not meet the assumption of normality, the Mann–Whitney U test was applied. To account for the risk of type I error due to multiple comparisons, Bonferroni correction was applied. A total of six predefined comparisons were made between the delirium and non-delirium groups, including four individual fatty acids (DGLA, AA, EPA, DHA) and two fatty acid ratios (AA/DGLA and EPA/AA). Simple linear regression was employed to assess relationships between pairs of continuous variables. To determine independent predictors of delirium in the CICU, multivariate logistic regression analysis was performed. Covariates were selected based on clinical relevance and previously established associations with the outcome. Variables showing significant associations in univariate analyses (*p* < 0.05) were considered for inclusion in the multivariate model, while avoiding multicollinearity. All statistical analyses were performed using JMP (version 17.0 for Macintosh, SAS Institute, Cary, NC, USA), and a two-tailed *p* value of <0.05 was considered statistically significant.

## 3. Results

### 3.1. Baseline Characteristics

A total of 590 patients with acute cardiovascular disease who were admitted to the CICU were included in this study. The average age of the participants was 70 years, with a standard deviation of 14 years. Among the enrolled patients, 400 (67.8%) were male, and all individuals were of Japanese ethnicity.

Delirium developed in 55 patients (9.3%). Age and percentage of men were significantly higher, and BMI was significantly lower in the delirium group. Comorbidities on admission, including atrial fibrillation and dementia, were significantly more common in the delirium group. Compared to the non-delirium group, the delirium group had a significantly higher incidence of ADHF and a significantly lower incidence of ACS as the reason for hospital admission. TC, LDL-C, HDL-C, and albumin levels were significantly lower in the delirium group. Serum creatinine, CRP, and NT-pro BNP levels were significantly higher in the delirium group. The use of antipsychotics was significantly higher in the delirium group, whereas there were no significant differences in the use of other medications between the two groups (Table 1). Additionally, the length of stay in the CICU was significantly longer in patients with delirium than in those without delirium (15.8 ± 19.0 vs. 6.4 ± 5.9 days, *p* < 0.001).

### 3.2. PUFA Levels and the Development of Delirium

As shown in Figure 1, DGLA levels were significantly lower in patients with delirium than in those without delirium (23.0 ± 10.2 vs. 31.4 ± 12.6 µg/mL, *p* < 0.001). In contrast, levels of AA (158.9 ± 60.8 vs. 168.7 ± 45.6 µg/mL, *p* = 0.053), EPA (50.1 ± 31.7 vs. 57.9 ± 38.8 µg/mL, *p* = 0.156), and DHA (120.9 ± 48.6 vs. 122.8 ± 43.7 µg/mL, *p* = 0.612) did not significantly differ between the two groups. Figure 2 shows that delta-5 desaturase activity estimated as AA/DGLA significantly increased in the delirium group than in the non-delirium group (7.4 ± 2.4 vs. 5.9 ± 2.0, *p* < 0.001). The EPA/AA ratio was not associated with the occurrence of delirium (0.34 ± 0.25 vs. 0.36 ± 0.26, *p* = 0.446). The *p*-values for DGLA and the AA/DGLA ratio were both below 0.001. After Bonferroni correction for multiple comparisons, these associations remained statistically significant (adjusted *p* < 0.006).

### 3.3. PUFA Levels, the ICDSC Score, and the Duration of Delirium

As shown in Table 2, the maximum delirium score showed weak but significant negative relationships with DGLA and AA, and positive relationships with delta-5 desaturase activity (DGLA: *p* < 0.001; AA: *p* = 0.002; AA/DGLA: *p* < 0.001). DGLA and AA levels were negatively associated with the duration of delirium (DGLA, *p* = 0.001; AA, *p* = 0.004). In addition, EPA and DHA levels tended to be negatively associated, and delta-5 desaturase activity (AA/DGLA) tended to be positively associated with the duration of delirium.

### 3.4. Multivariate Analysis

After controlling for potential confounders, both DGLA and delta-5 desaturase activity (AA/DGLA) were independent risk factors for onset of delirium in the CICU (Table 3). Model 1 was adjusted for age, sex, and BMI; Model 2 was adjusted for Model 1 and presence of ADHF and dementia; and Model 3 was adjusted for Model 2 and inflammation (CRP levels), renal dysfunction (creatinine levels), and malnutrition (albumin levels).

## 4. Discussion

This study revealed a significant association between reduced omega-6 PUFA levels at the time of admission and the occurrence of delirium in patients with acute cardiovascular disease admitted to the CICU. Specifically, lower levels of DGLA and elevated delta-5 desaturase activity (reflected by the AA/DGLA ratio) were independently linked to the presence of delirium. These findings suggest that diminished omega-6 PUFAs—particularly DGLA—may serve as potential markers for identifying patients at elevated risk for developing delirium. To our knowledge, this is the first investigation to demonstrate a relationship between low omega-6 PUFA levels and delirium onset.

The biological processes contributing to the development of delirium are not yet fully elucidated [1]. Nonetheless, growing evidence indicates that delirium arises from the complex interplay of multiple biological processes, ultimately disturbing large-scale neural networks and resulting in acute cognitive impairment. Proposed mechanisms underlying this condition include imbalances in neurotransmitter systems, systemic inflammation, physical stress, metabolic abnormalities, disturbances in electrolyte balance, and genetic predispositions [3]. However, the concept of a singular, unified mechanism—often referred to as the “final common pathway”—for delirium remains unresolved [3,24].

Several studies have investigated the relationship between PUFAs and cognitive impairments. One such study suggested that long-chain omega-3 PUFAs may enhance cerebral function by promoting vasodilation and improving blood flow through their effects on endothelial cells within the brain’s vascular system [25]. Another investigation found that a higher dietary omega-6 to omega-3 ratio was positively correlated with an increased risk of developing Alzheimer’s disease [26]. A recent study offered new insights into how a balanced ratio of plasma phospholipid omega-3 and omega-6 PUFAs may support memory function and maintain the integrity of white matter microstructure [27]. However, in a previous study, intravenous delivery of omega-3 fatty acids in critically ill patients with sepsis did not reduce the incidence of septic-associated delirium [28].

This study found that elevated delta-5 desaturase activity, potentially leading to decreased DGLA and increased AA levels, was linked to the onset of delirium. Delta-5 desaturase introduces a double bond at the fifth position from the carboxyl end of fatty acid chains. Through this enzymatic activity, it converts DGLA into AA and eicosatetraenoic acid into EPA, using these molecules as its respective substrates [29]. Recent studies have linked delta-5 desaturase activity to various health conditions, such as cardiovascular disease, diabetes, cancer, and neurological disorders [30,31,32,33]. A study suggested that gene expression of delta-5 desaturase influences cognitive function across diverse neural tissues and distinct cell populations [33]. However, how delta-5 desaturase activity influences brain function remains unclear, and further studies are warranted to clarify the mechanisms underlying these associations.

Growing evidence demonstrates the coronary artery disease (CAD) benefits of dietary omega-6 PUFAs [34]. Replacing saturated fatty acids (SFAs) with omega-6 PUFAs lowered the incidence of CAD events by 24% in a pooled analysis of six randomized controlled trials, an effect likely mediated by substantial lowering of LDL-C. Additionally, a substantial body of evidence from short-term controlled feeding trials has shown that omega-6 polyunsaturated fatty acids possess cholesterol-lowering effects independent of simply replacing saturated fats [35,36,37]. In our previous work, we found that reduced levels of omega-6 fatty acids were linked to unfavorable outcomes in patients with acute cardiovascular conditions and ADHF [19,23]. There is still controversy surrounding the mechanisms by which omega-6 PUFAs may contribute to cardiovascular disease through their potential role in delirium development. Therefore, future studies should focus on the associations between PUFAs and cardiovascular disease and cognitive function.

The 2013 guidelines on lifestyle management to reduce cardiovascular risk suggest the replacement of PUFAs with SFA [38]. Furthermore, the 2019 ACC/AHA Guideline on the primary prevention of cardiovascular disease recommends limiting trans fat intake due to its harmful effects on lipid profiles and lipoproteins, as well as its role in promoting endothelial dysfunction, insulin resistance, inflammation, and arrhythmias. These effects have also been linked to increased all-cause mortality in the REGARDS study [39,40]. However, there is no evidence to support that omega-6 PUFA supplementation improves the prognosis of patients with cardiovascular diseases. Further studies are required to establish dietary guidelines, particularly concerning the optimal fatty acid composition for patients with cardiovascular disease.

This study has several limitations. First, it was conducted at a single center with a relatively small sample size. Larger, multicenter studies are warranted to more accurately evaluate the relationship between various nutritional indicators, PUFA levels, and delirium onset in patients with cardiovascular disease. Second, we were unable to obtain detailed information on patients’ dietary habits and physical activity levels. As a result, we could not explore potential links between pre-admission lifestyle factors and PUFA status. Third, although the number of primary comparisons was limited and Bonferroni correction was applied, multiple statistical tests were conducted, which increases the risk of type I error—namely, observing statistically significant results by chance despite the absence of a true effect. Therefore, our findings should be interpreted with appropriate caution.

## 5. Conclusions

Low levels of omega-6 PUFAs, particularly DGLA, at the time of admission were significantly associated with the onset of delirium in patients with acute cardiovascular disease admitted to the CICU. These findings suggest that reduced omega-6 PUFA levels may serve as a potential marker for identifying patients at elevated risk of developing delirium. Further studies are needed to elucidate the biological pathways that link PUFA status to delirium. From a public health standpoint, attention is increasingly being directed toward the balance between omega-6 and omega-3 fatty acids. Consequently, identifying the determinants of this PUFA ratio is highly relevant for advancing cardiovascular health strategies. Ultimately, we aim to contribute to the development of practical dietary and clinical protocols that support better outcomes in patients with cardiovascular disease.

## Figures and Tables

**Figure 1 nutrients-17-01979-f001:**
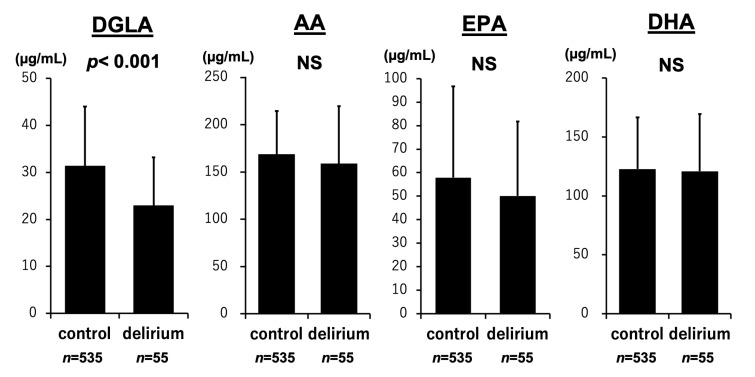
PUFAs and development of delirium. A comparison of PUFA concentrations revealed that patients with delirium had markedly lower levels of DGLA than those without delirium. Conversely, no significant differences were found between the two groups in terms of AA, EPA, or DHA levels. Results are expressed as mean ± standard deviation.

**Figure 2 nutrients-17-01979-f002:**
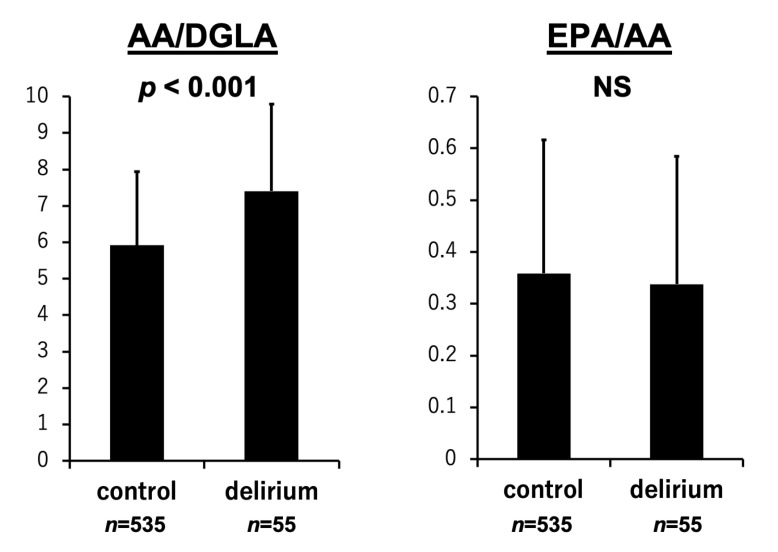
Balance of PUFAs and development of delirium. The AA/DGLA ratio, which reflects delta-5 desaturase activity, was significantly higher in patients who developed delirium than in those who did not. In contrast, no significant relationship was observed between the EPA/AA ratio and the incidence of delirium. Results are expressed as mean ± standard deviation.

**Table 1 nutrients-17-01979-t001:** Baseline characteristics of the study participants on admission (*n* = 590).

	Delirium Group (*n* = 55)	Non-Delirium Group (*n* = 535)	*p*
Age, years	80.3 ± 10.9	69.0 ± 13.8	<0.001
Male, *n* (%)	28 (51)	372 (70)	0.01
Body mass index, kg/m^2^	22.4 ±3.9	23.7 ± 4.5	0.02
Left ventricular ejection fraction, %	55 ± 14	55 ± 16	0.70
Comorbidities on admission			
Dementia, *n* (%)	16 (29)	11 (2)	<0.001
Atrial fibrillation, *n* (%)	15 (28)	83 (16)	0.03
Malignancy, *n* (%)	11 (20)	59 (11)	0.05
Hypertension, *n* (%)	35 (64)	279 (52)	0.10
Diabetes mellitus, *n* (%)	23 (42)	154 (29)	0.12
Cerebral infarction, *n* (%)	6 (11)	42 (7.9)	0.43
Reason of hospital admission			
Acute decompensated heart failure, *n* (%)	35 (64)	204 (38)	<0.001
Acute coronary syndrome, *n* (%)	7 (13)	185 (35)	0.001
Aortic disease, *n* (%)	4 (7.3)	16 (3.0)	0.10
VT/VF, *n* (%)	1 (1.8)	15 (2.8)	1.00
PTE/DVT, *n* (%)	2 (3.6)	20 (3.7)	1.00
Others, *n* (%)	6 (11)	95 (18)	0.54
Laboratory data			
Albumin, g/dL	3.13 ± 0.57	3.47 ± 0.56	<0.001
CRP, mg/dL	4.3 ± 6.0	1.9 ± 3.5	<0.001
NT-pro BNP, pg/mL	14718 ± 22640	6202 ± 15499	<0.001
Total cholesterol, mg/dL	147.8 ± 50.5	163.0 ± 40.4	0.002
Creatinine, mg/dL	1.92 ± 2.22	1.44 ± 1.81	0.003
LDL-C, mg/dL	87.6 ± 34.2	100.7 ± 33.3	0.003
HDL-C, mg/dL	39.1 ± 12.5	43.3 ± 13.8	0.02
Triglycerides, mg/dL	86.0 ± 57.0	93.3 ± 52.7	0.11
HbA1c, %	6.43 ± 2.06	6.16 ± 0.99	0.51
Medication			
Antipsychotics, *n* (%)	2 (3.7)	2 (0.4)	0.04
Anticoagulants, *n* (%)	14 (26)	89 (17)	0.10
Nonbenzodiazepines, *n* (%)	2 (3.7)	6 (1.1)	0.16
Antiplatelets, *n* (%)	16 (29)	194 (36)	0.29
β-blockers, *n* (%)	20 (36)	169 (32)	0.50
Calcium channel blockers, *n* (%)	15 (27)	168 (31)	0.53
Insulin, *n* (%)	4 (7)	30 (6)	0.55
Statin, *n* (%)	16 (29)	176 (33)	0.57
Oral hypoglycemic agents, *n* (%)	11 (20)	93 (18)	0.59
ACE-I/ARBs, *n* (%)	20 (36)	178 (33)	0.64
Anti-depressants, *n* (%)	0 (0)	3 (0.6)	1.0
Anxiolytic drugs, *n* (%)	0 (0)	8 (1.5)	1.0
Benzodiazepines, *n* (%)	1 (1.9)	19 (3.6)	1.0

Data are presented as means ± SD or number (percentage). VT, ventricular tachycardia; VF, ventricular fibrillation; PTE, pulmonary thromboembolism; DVT, deep vein thrombosis; CRP, C-reactive protein; NT-pro BNP, *N*-terminal pro brain natriuretic peptide; LDL-C, low-density lipoprotein cholesterol; HDL-C, high-density lipoprotein cholesterol; HbA1c, hemoglobin A1c, national glycohemoglobin standardization program calculation; ACE-I, angiotensin-converting-enzyme inhibitor; ARBs, angiotensin II receptor blockers.

**Table 2 nutrients-17-01979-t002:** PUFA levels and the ISDSC score, PUFA levels and the duration of delirium.

	Maximum ISDSC Score		Duration of Delirium	
	r	*p*	r	*p*
DGLA (mg/dL)	**−0.231**	**<0.001**	**−0.134**	**0.001**
AA (mg/dL)	**−0.131**	**0.002**	**−0.120**	**0.004**
EPA (mg/dL)	−0.064	0.120	−0.080	0.052
DHA (mg/dL)	−0.048	0.236	−0.079	0.055
EPA/AA	−0.001	0.976	−0.031	0.456
AA/DGLA	**0.168**	**<0.001**	0.069	0.093

The bold means statistically significant values. DGLA, dihomo-gamma-linolenic acid; AA, arachidonic acid; EPA, eicosapentaenoic acids; DHA, docosahexaenoic acid.

**Table 3 nutrients-17-01979-t003:** Multivariate logistic regression analyses for the occurrence of delirium.

	Univariate			Model 1			Model 2			Model 3		
	OR	95%CI	*p*	OR	95%CI	*p*	OR	95%CI	*p*	OR	95%CI	*p*
DGLA, 1 increase	**0.92**	**0.90** **–0.96**	**<0.001**	**0.93**	**0.90–0.97**	**<0.001**	**0.94**	**0.90–0.97**	**0.001**	**0.95**	**0.91–0.99**	**0.009**
DGLA as a categorical variable												
1st quartile	1.00	reference		1.00	reference		1.00	reference		1.00	reference	
2nd quartile	**0.30**	**0.14** **–0.62**	**0.001**	**0.33**	**0.15** **–0.70**	**0.004**	**0.30**	**0.13** **–0.69**	**0.005**	**0.35**	**0.15** **–0.83**	**0.02**
3rd quartile	**0.22**	**0.10** **–0.49**	**<0.001**	**0.23**	**0.10** **–0.55**	**<0.001**	**0.20**	**0.08** **–0.52**	**0.001**	**0.26**	**0.09** **–0.70**	**0.008**
4th quartile	**0.13**	**0.05–0.35**	**<0.001**	**0.20**	**0.07–0.57**	**0.003**	**0.21**	**0.07** **–0.62**	**0.005**	**0.22**	**0.06** **–0.79**	**0.02**
AA/DGLA, 1 increase	**1.33**	**1.18–1.50**	**<0.001**	**1.29**	**1.14–1.47**	**<0.001**	**1.30**	**1.13–1.49**	**<0.001**	**1.28**	**1.11–1.48**	**<0.001**
AA/DGLA as a categorical variable												
1st quartile	1.00	reference		1.00	reference		1.00	reference		1.00	reference	
2nd quartile	2.28	0.77–6.73	0.14	1.62	0.53–4.95	0.40	1.78	0.56–5.70	0.33	2.20	0.62–7.80	0.22
3rd quartile	2.52	0.87–7.36	0.09	1.51	0.50–4.59	0.46	1.27	0.39–4.08	0.69	1.32	0.36–4.78	0.67
4th quartile	**6.34**	**2.37** **–17.0**	**<0.001**	**3.90**	**1.40–10.9**	**0.009**	**3.64**	**1.24–10.7**	**0.02**	**3.74**	**1.12–12.5**	**0.03**

Model 1 was adjusted for age, sex, and BMI; Model 2 was adjusted for Model 1 and presence of ADHF and dementia; and Model 3 was adjusted for Model 2 and inflammation (C-reactive protein levels), renal dysfunction (creatinine levels), and malnutrition (albumin levels), OR, odds ratio; 95%CI, 95% confidence interval. The bold means statistically significant values. DGLA, dihomo-gamma-linolenic acid; AA, arachidonic acid.

## Data Availability

The original contributions presented in the study are included in the article, further inquiries can be directed to the corresponding author.

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
