# Peer review of "Association Between Low Omega-6 Polyunsaturated Fatty Acid Levels and the Development of Delirium in the Coronary Intensive Care Unit"

_nutrients, 2025, doi:10.3390/nu17121979_

Round 1
Reviewer 1 Report
Comments and Suggestions for Authors
The title and the introduction represent the content of the article very well.
There are some important confounding factors missing: had the length of the stay in the CICU an effect? Was there a presence or absence of factors which could provide some orientation for the patient in time (date and hour), place and persons (pictures of relatives). Was there a maintaince of day-night rythm? Was there an effect of sleep disturbance?
Table 1 should be improved by listing the factors according the p-value (the most significant on top). This identifis age and dementia as the most significant ones. This will also demonstrate more easily that antipsychotic medication is the only one with an effect.
Author Response
We sincerely thank Reviewer 1 for the thoughtful and constructive comments. We carefully considered each point and have revised the manuscript accordingly. Our detailed responses are provided below.
- There are some important confounding factors missing: had the length of the stay in the CICU an effect?
Author’s response: We sincerely thank the reviewer for this insightful comment. Although the length of stay in the CICU was significantly longer in patients with delirium compared to those without (15.8 ± 19.0 vs. 6.4 ± 5.9 days, P < 0.001), this variable is not available at the time of admission and thus cannot be considered a predictive factor. Moreover, a causal relationship cannot be determined, as the onset of delirium may have contributed to prolonged CICU stays. In light of this, we have chosen to report CICU stay duration in the Results section only, without including it as an explanatory variable in the predictive analysis.
Result section (Page 4, Line 191)
The use of antipsychotics was significantly higher in the delirium group, whereas there were no significant differences in the use of other medications between the two groups (Table 1). Additionally, the length of stay in the CICU was significantly longer in patients with delirium than in those without delirium (15.8 ± 19.0 vs. 6.4 ± 5.9 days, P < 0.001). Additionally, the length of stay in the CICU was significantly longer in patients with delirium than in those without delirium (15.8 ± 19.0 vs. 6.4 ± 5.9 days, P < 0.001).
- Was there a presence or absence of factors which could provide some orientation for the patient in time (date and hour), place and persons (pictures of relatives).
Author’s response: We sincerely thank the reviewer for their valuable suggestion. To help orient patients to time and place, each room was equipped with a clock, and nurses routinely confirmed the date with patients each day. Bringing photographs of family members was also permitted. We have added the following sentence to the Methods section accordingly.
Methods section (Page 3, Line 107)
Delirium was identified based on an Intensive Care Delirium Screening Checklist (ICDSC) score of 4 or higher [20]. The evaluations were performed three times daily by independent nursing staff. To support the circadian rhythm, patients in the CICU were exposed to natural daylight during the day, and the lights were dimmed at night to simulate a normal day-night cycle. Additionally, to help orient patients to time and place, each room was equipped with a clock, and nurses routinely confirmed the date with patients each day. Bringing photographs of family members was also permitted.
- Was there a maintained of day-night rythm?
Author’s response: We sincerely thank the reviewer for their valuable suggestion. To support the circadian rhythm, patients in our CICU were exposed to natural daylight during the day, and the lights were dimmed at night to simulate a normal day-night cycle. We have added the following sentence to the Methods section accordingly.
Methods section (Page 3, Line 107)
Delirium was identified based on an Intensive Care Delirium Screening Checklist (ICDSC) score of 4 or higher [20]. The evaluations were performed three times daily by independent nursing staff. To support the circadian rhythm, patients in the CICU were exposed to natural daylight during the day, and the lights were dimmed at night to simulate a normal day-night cycle. Additionally, to help orient patients to time and place, each room was equipped with a clock, and nurses routinely confirmed the date with patients each day. Bringing photographs of family members was also permitted.
- Was there an effect of sleep disturbance?
Author’s response:This study was retrospective in nature, and unfortunately, the presence or absence of sleep disorders could not be determined. However, as shown in Table 1, there were no statistically significant differences in the use of benzodiazepines or non-benzodiazepine hypnotics between patients with and without delirium.
Reviewer 2 Report
Comments and Suggestions for Authors
The authors present an analysis of omega levels and delirium in the ICU. A few points for consideration.
-I would encourage the authors to make their introduction more comprehensive. There needs to be additional information for a reader on the omega/PUFA pathway, how it works and definitions, how diet can influence it and its effects on the brain. This would be highly useful to clinicians reading your article and interested in understand what mechanisms and physiology link omega/PUFA to the outcome you are studying.
-please give a full list of inclusion and exclusion criteria for your sample set.
-Are delirium scores completed on all 653 enrolled patients? who is doing the rating?
-Were other demographic or background information collected such as sex, medication status, reason for admission, etc? There appears to be more like BMI which is not listed in your methods.
-can you give some context for the definitions being listed starting at line 84? This seems to just come out of nowhere and could be written with a narrative sense.
-where were all the labs analyzed? In the hospital or at an external site that does lipid analysis?
-The complete set of metabolites that were analyzed should be included in the methods. Your abstract mentions AA and DGLA which is not in the methods. What exactly was measured and how?
-was normality assessed for variables in your statistical analysis? Please give a better rationale or definition of how and what you chose to include in your regressions for controlling.
-Please list the actual p-values instead of "NS".
-Why weren't comparisons made for diagnosis upon admission? This may be important because it could influence your analyses?
-Although the figures and other tables are useful, please list your PUFA metabolites in Table 1 under laboratory data so that we can get a sense of the variability of each data point.
-You have to at least list multiple testing and not controlling for type I error in your limitation section since you have done a lot of statistical tests.
Author Response
We sincerely thank Reviewer 2 for the thoughtful and constructive comments. We carefully considered each point and have revised the manuscript accordingly. Our detailed responses are provided below.
The authors present an analysis of omega levels and delirium in the ICU. A few points for consideration.
- I would encourage the authors to make their introduction more comprehensive. There needs to be additional information for a reader on the omega/PUFA pathway, how it works and definitions, how diet can influence it and its effects on the brain. This would be highly useful to clinicians reading your article and interested in understand what mechanisms and physiology link omega/PUFA to the outcome you are studying.
Author’s response:We are grateful to the reviewer for their valuable suggestion. Accordingly, we have revised the Introduction to include a brief overview of the physiological roles of omega fatty acids and their relevance to disease pathogenesis.
Introduction section (Page 2, Line 63)
Essential fatty acids (EFAs) and their long-chain polyunsaturated derivatives (LCPs) play a fundamental role in human growth and health maintenance. Since humans lack the ability to produce EFAs and can only inefficiently convert them into LCPs, daily dietary intake of EFAs is essential [8, 9]. The primary EFAs in the n-6 and n-3 families are linoleic acid (LA; 18:2n-6) and alpha-linolenic acid (ALA; 18:3n-3), respectively. These parent fatty acids undergo desaturation and elongation in the body, leading to the formation of longer-chain, more unsaturated LCPs. Among the metabolites of LA, dihomo-gamma-linolenic acid (DGLA; 20:3n-6) and arachidonic acid (AA; 20:4n-6) are particularly notable. These are classified as omega-6 polyunsaturated fatty acids (PUFAs) due to the location of their first double bond at the sixth carbon from the methyl end. On the other hand, eicosapentaenoic acid (EPA; 20:5n-3) and docosahexaenoic acid (DHA; 22:6n-3), which are generated from ALA, belong to the omega-3 PUFA family, as their first double bond appears at the third carbon [10, 11]. Omega-3 PUFAs are predominantly found in marine oils, such as fish oil, whereas omega-6 PUFAs are abundant in meats and in plant oils like sunflower, safflower, and corn oil [12].
In recent years, nutritional assessment has become an increasingly recognized component of cardiovascular disease management [13]. Of particular interest is the role of PUFAs in cardiovascular health, which has been the subject of extensive investigation [14-18]. In our earlier research, we found that reduced levels of omega-6 polyunsaturated fatty acids were significantly linked to increased long-term mortality across different nutritional statuses in patients with acute decompensated heart failure (ADHF) [19]. There is still a lack of evidence identifying which types of PUFAs may help reduce the incidence of delirium and improve prognosis in individuals with heart disease.
- Please give a full list of inclusion and exclusion criteria for your sample set.
Author’s response:Thank you for your valuable comment. In response, we have provided a full list of the inclusion and exclusion criteria for our study population as follows.
Inclusion criteria:
- Patients consecutively admitted to the CICU of Juntendo University Hospital between January 2015 and December 2016
- Availability of both delirium assessment and PUFA measurements
Exclusion criteria:
- Repeat CICU admissions during the study period (only the first admission was included; n = 85)
- Incomplete delirium score assessments (n = 36)
- Requirement for respiratory support or severe disturbance of consciousness, which made delirium assessment difficult (n = 9)
- Missing PUFA measurements (n = 63)
As a result, 590 patients with complete data were included in the final analysis. Based on these criteria, we have described the process of patient enrollment in the Methods section.
Methods section (Page 2, Line 93)
This investigation was embedded within an ongoing prospective cohort study focusing on biomarkers in CICU-admitted patients. Although the hypothesis concerning PUFAs emerged retrospectively, all data were gathered in a systematic and prospective manner. We screened 783 consecutive patients who were admitted to the CICU of Juntendo University Hospital between January 2015 and December 2016. Among patients with multiple CICU admissions during the study period, only the first admission was included; subsequent admissions (n = 85) were excluded. Additional 36 patients were excluded due to incomplete delirium score assessments. Because of the difficulty in evaluating delirium, 9 patients who either required mechanical ventilation or exhibited profound impairments in consciousness were also excluded. Furthermore, 63 patients without PUFA measurements were excluded. As a result, 590 patients with both completed delirium assessments and PUFA level measurements were included in the final analysis. These patients were divided into two groups—delirium and non-delirium—to investigate the association between PUFA levels and the occurrence of delirium.
- Are delirium scores completed on all 653 enrolled patients? who is doing the rating?
Author’s response:We sincerely thank the reviewer for this important question. As described in the Methods section, all 653 enrolled patients underwent delirium assessment. However, the process by which 590 patients were ultimately included in the final analysis was not clearly described in the original manuscript. Therefore, we have now provided a more detailed explanation of the patient selection process in the revised Methods section. Delirium assessments were performed three times daily by independent nursing staff, and this has also been added to the Methods section accordingly.
Methods section (Page 2, Line 93)
This investigation was embedded within an ongoing prospective cohort study focusing on biomarkers in CICU-admitted patients. Although the hypothesis concerning PUFAs emerged retrospectively, all data were gathered in a systematic and prospective manner. We screened 783 consecutive patients who were admitted to the CICU of Juntendo University Hospital between January 2015 and December 2016. Among patients with multiple CICU admissions during the study period, only the first admission was included; subsequent admissions (n = 85) were excluded. Additional 36 patients were excluded due to incomplete delirium score assessments. Because of the difficulty in evaluating delirium, 9 patients who either required mechanical ventilation or exhibited profound impairments in consciousness were also excluded. Furthermore, 63 patients without PUFA measurements were excluded. As a result, 590 patients with both completed delirium assessments and PUFA level measurements were included in the final analysis. These patients were divided into two groups—delirium and non-delirium—to investigate the association between PUFA levels and the occurrence of delirium.
Methods section (Page 3, Line 107)
Delirium was identified based on an Intensive Care Delirium Screening Checklist (ICDSC) score of 4 or higher [20]. The evaluations were performed three times daily by independent nursing staff. To support the circadian rhythm, patients in the CICU were exposed to natural daylight during the day, and the lights were dimmed at night to simulate a normal day-night cycle.
- Were other demographic or background information collected such as sex, medication status, reason for admission, etc? There appears to be more like BMI which is not listed in your methods.
Author’s response:We greatly appreciate the reviewer’s insightful comment. As indicated, information such as sex, medication status, reason for admission, and BMI is already presented in the Table 1. To improve transparency, we have now included a description of these variables in the Methods section as well.
Methods section (Page 3, Line 114)
All relevant data were collected from medical records at the time of hospital admission, including age, sex, body mass index (BMI), left ventricular ejection fraction (LVEF), comorbidity history (e.g., diabetes mellitus, dyslipidemia, hypertension, atrial fibrillation, dementia, cerebral infarction, and malignancy), reason for admission (e.g., ADHF, acute coronary syndrome (ACS), aortic dissection, pulmonary thromboembolism/deep vein thrombosis, ventricular tachycardia/ventricular fibrillation, and other causes), laboratory test results, and pre-admission medication use.
To ensure consistency and reproducibility, the definitions of key clinical conditions at the time of hospital admission used in this study are provided as follows. BMI was measured at the time of admission. LVEF was measured by transthoracic echocardiography upon admission. Diabetes mellitus was determined by either a documented diagnosis in the admission records, a hemoglobin A1c (HbA1c) level exceeding 6.5% as standardized by the National Glycohemoglobin Standardization Program, or the use of antidiabetic medications, including insulin. Dyslipidemia was identified through a recorded diagnosis at admission, abnormal lipid levels—specifically triglycerides (TG) >150 mg/dL, low-density lipoprotein cholesterol (LDL-C) >140 mg/dL, or high-density lipoprotein cholesterol (HDL-C) <40 mg/dL—or current use of lipid-lowering therapy. Hypertension was defined as a diagnosis noted at admission, systolic blood pressure greater than 140 mmHg and/or diastolic pressure over 90 mmHg, or ongoing treatment with antihypertensive drugs. Atrial fibrillation, dementia, prior cerebral infarction, and malignancy were confirmed based on documented admission diagnoses in the medical records.
The definitions of reasons for hospital admission were as follows. The diagnosis of ADHF was based on the criteria established by the Framingham study. [21]. ACS was characterized by the presence of unstable angina, non–ST-elevation myocardial infarction, or ST-elevation myocardial infarction [22]. Aortic diseases, pulmonary thromboembolism/deep vein thrombosis, ventricular tachycardia/ventricular fibrillation, and other conditions were defined based on the documented admission diagnoses in the medical records.
- Can you give some context for the definitions being listed starting at line 84? This seems to just come out of nowhere and could be written with a narrative sense.
Author’s response:We sincerely thank the reviewer for this insightful comment. In response, we have revised the manuscript to provide appropriate context before introducing the definitions starting at line 84. Specifically, we have added a sentence explaining that these definitions are based on clinical data collected at the time of hospital admission, in order to ensure consistency and reproducibility in our analysis. This revision aims to improve the narrative flow and clarity of the Methods section.
Methods section (Page 3, Line 114)
All relevant data were collected from medical records at the time of hospital admission, including age, sex, body mass index (BMI), left ventricular ejection fraction (LVEF), comorbidity history (e.g., diabetes mellitus, dyslipidemia, hypertension, atrial fibrillation, dementia, cerebral infarction, and malignancy), reason for admission (e.g., ADHF, acute coronary syndrome (ACS), aortic dissection, pulmonary thromboembolism/deep vein thrombosis, ventricular tachycardia/ventricular fibrillation, and other causes), laboratory test results, and pre-admission medication use.
To ensure consistency and reproducibility, the definitions of key clinical conditions at the time of hospital admission used in this study are provided as follows. BMI was measured at the time of admission. LVEF was measured by transthoracic echocardiography upon admission. Diabetes mellitus was determined by either a documented diagnosis in the admission records, a hemoglobin A1c (HbA1c) level exceeding 6.5% as standardized by the National Glycohemoglobin Standardization Program, or the use of antidiabetic medications, including insulin. Dyslipidemia was identified through a recorded diagnosis at admission, abnormal lipid levels—specifically triglycerides (TG) >150 mg/dL, low-density lipoprotein cholesterol (LDL-C) >140 mg/dL, or high-density lipoprotein cholesterol (HDL-C) <40 mg/dL—or current use of lipid-lowering therapy. Hypertension was defined as a diagnosis noted at admission, systolic blood pressure greater than 140 mmHg and/or diastolic pressure over 90 mmHg, or ongoing treatment with antihypertensive drugs. Atrial fibrillation, dementia, prior cerebral infarction, and malignancy were confirmed based on documented admission diagnoses in the medical records.
The definitions of reasons for hospital admission were as follows. The diagnosis of ADHF was based on the criteria established by the Framingham study. [21]. ACS was characterized by the presence of unstable angina, non–ST-elevation myocardial infarction, or ST-elevation myocardial infarction [22]. Aortic diseases, pulmonary thromboembolism/deep vein thrombosis, ventricular tachycardia/ventricular fibrillation, and other conditions were defined based on the documented admission diagnoses in the medical records.
- Where were all the labs analyzed? In the hospital or at an external site that does lipid analysis? The complete set of metabolites that were analyzed should be included in the methods. Your abstract mentions AA and DGLA which is not in the methods. What exactly was measured and how?
Author’s response:We thank the reviewer for their helpful and constructive comment. In response, we have added detailed information regarding the measurement of polyunsaturated fatty acids (PUFAs), including arachidonic acid (AA) and dihomo-γ-linolenic acid (DGLA), to the Methods section. The measurement methods and the responsible laboratory are now clearly described to enhance reproducibility and clarity.
Methods section (Page 3, Line 149)
PUFAs, including DGLA, AA, EPA, and DHA, were measured using gas chromatography by SRL, Inc. (Tokyo, Japan), a certified clinical laboratory. All other laboratory tests were performed at the hospital’s central laboratory using standardized procedures.
- Was normality assessed for variables in your statistical analysis? Please give a better rationale or definition of how and what you chose to include in your regressions for controlling.
Author’s response: We thank the reviewer for their valuable comment. As noted, we had already assessed the normality of continuous variables using the Shapiro–Wilk test. For variables that did not meet the assumption of normality, we appropriately used non-parametric tests such as the Mann–Whitney U test. We have corrected the Methods section accordingly to accurately reflect these statistical procedures.
Regarding the regression analysis, we selected covariates based on clinical relevance and previously established associations with the outcome of interest. Variables with significant associations in univariate analyses (p < 0.05) were further considered for inclusion in the multivariate models, while also avoiding multicollinearity. This approach ensured appropriate adjustment for potential confounders. These details have also been added to the Methods section to enhance clarity and reproducibility.
Methods section (Page 4, Line 159)
Continuous data are presented as means with standard deviations, while categorical data are shown as percentages. Group comparisons were conducted using Student’s t-test, the chi-square test, or Fisher's exact test, as appropriate. The distribution of continuous variables was evaluated, and normality was assessed using the Shapiro–Wilk test. For variables that did not meet the assumption of normality, the Mann–Whitney U test was applied. To account for the risk of type I error due to multiple comparisons, Bonferroni correction was applied. A total of six predefined comparisons were made between the delirium and non-delirium groups, including four individual fatty acids (DGLA, AA, EPA, DHA) and two fatty acid ratios (AA/DGLA and EPA/AA). Simple linear regression was employed to assess relationships between pairs of continuous variables. To determine independent predictors of delirium in the CICU, multivariate logistic regression analysis was performed. Covariates were selected based on clinical relevance and previously established associations with the outcome. Variables showing significant associations in univariate analyses (p < 0.05) were considered for inclusion in the multivariate model, while avoiding multicollinearity.
- Please list the actual p-values instead of "NS".
Author’s response:We appreciate the reviewer’s valuable comment. In response, we have revised the tables to provide the actual p-values for all statistical comparisons, including those that were not statistically significant. This change improves transparency and allows for a more accurate interpretation of the results.
The revised tables have been updated accordingly and now include all actual p-values. Please refer to the new versions.
- Why weren't comparisons made for diagnosis upon admission? This may be important because it could influence your analyses?
Author’s response:We thank the reviewer for their insightful comment. In the original analysis, we compared the overall distribution of admission diagnoses between groups using the chi-square test. However, we did not initially examine the association between each specific diagnosis and the occurrence of delirium. In response to the reviewer’s suggestion, we reanalyzed the data to assess the relationship between individual admission diagnoses and the incidence of delirium. As shown in the revised manuscript and Table 1, a higher prevalence of acute decompensated heart failure (ADHF) (P < 0.001) and a lower prevalence of acute coronary syndrome (ACS) (P = 0.001) were significantly associated with the occurrence of delirium. Based on these findings, ADHF was included as an independent variable in the multivariate logistic regression model to further evaluate its potential role as a risk factor.
Result section (Page 4, Line 185)
Comorbidities on admission, including atrial fibrillation and dementia, were significantly more common in the delirium group. Compared to the non-delirium group, the delirium group had a significantly higher incidence of ADHF and a significantly lower incidence of ACS as the reason for hospital admission.
- Although the figures and other tables are useful, please list your PUFA metabolites in Table 1 under laboratory data so that we can get a sense of the variability of each data point.
Author’s response:We appreciate the reviewer’s thoughtful suggestion. While we agree that presenting PUFA metabolite values can enhance the reader’s understanding, we have chosen not to include them in Table 1 to avoid redundancy with the existing figure, which already illustrates the distribution and variability of these data in detail. 
Instead, we have added representative values (mean ± standard deviation) of the key PUFA metabolites to the Results section of the main text. We believe this approach maintains clarity while still providing the necessary numerical data to support interpretation.
Result section (Page 6, Line 209)
As shown in Figure 1, DGLA levels were significantly lower in patients with delirium than in those without delirium (23.0 ± 10.2 vs. 31.4 ± 12.6 µg/mL, P < 0.001). In contrast, levels of AA (158.9 ± 60.8 vs. 168.7 ± 45.6 µg/mL, P = 0.053), EPA (50.1 ± 31.7 vs. 57.9 ± 38.8 µg/mL, P = 0.156), and DHA (120.9 ± 48.6 vs. 122.8 ± 43.7 µg/mL, P = 0.612) did not significantly differ between the two groups. Figure 2 shows that delta-5 desaturase activity estimated as AA/DGLA significantly increased in the delirium group than in the non-delirium group (7.4 ± 2.4 vs. 5.9 ± 2.0, P < 0.001). The EPA/AA ratio was not associated with the occurrence of delirium (0.34 ± 0.25 vs. 0.36 ± 0.26, P = 0.446).
- You have to at least list multiple testing and not controlling for type I error in your limitation section since you have done a lot of statistical tests.
Author’s response:We thank the reviewer for this important suggestion. We acknowledge that multiple statistical tests were conducted in this study. However, the primary statistical comparisons were limited to six predefined tests: four individual fatty acids (DGLA, AA, EPA, DHA) and two fatty acid ratios (AA/DGLA and EPA/AA), comparing the delirium and non-delirium groups. The p-values for DGLA and the AA/DGLA ratio, which showed statistically significant differences, were both below 0.001. Even after applying Bonferroni correction for multiple comparisons, the adjusted p-values remained below 0.006. Therefore, these findings remain statistically significant. Nonetheless, we recognize that the risk of false-positive results cannot be completely eliminated, even with correction for multiple testing. This limitation has now been clearly stated in the revised manuscript.
Methods section (Page 4, Line 164)
To account for the risk of type I error due to multiple comparisons, Bonferroni correction was applied. A total of six predefined comparisons were made between the delirium and non-delirium groups, including four individual fatty acids (DGLA, AA, EPA, DHA) and two fatty acid ratios (AA/DGLA and EPA/AA).
Result section (Page 6, Line 216)
The P-values for DGLA and the AA/DGLA ratio were both below 0.001. After Bonferroni correction for multiple comparisons, these associations remained statistically significant (adjusted P < 0.006).
Limitations section (Page 11, Line 63)
This study has several limitations. First, it was conducted at a single center with a relatively small sample size. Larger, multicenter studies are warranted to more accurately evaluate the relationship between various nutritional indicators, PUFA levels, and delirium onset in patients with cardiovascular disease. Second, we were unable to obtain detailed information on patients’ dietary habits and physical activity levels. As a result, we could not explore potential links between pre-admission lifestyle factors and PUFA status. Third, although the number of primary comparisons was limited and Bonferroni correction was applied, multiple statistical tests were conducted, which increases the risk of type I error—namely, observing statistically significant results by chance despite the absence of a true effect. Therefore, our findings should be interpreted with appropriate caution.

Round 2
Reviewer 1 Report
Comments and Suggestions for Authors
The manuscript has been improved.
Reviewer 2 Report
Comments and Suggestions for Authors
Thank you for answer reviewer comments and revising your manuscript accordingly. I have no further comments for revisions.